# Krill Oil’s Protective Benefits against Ultraviolet B-Induced Skin Photoaging in Hairless Mice and In Vitro Experiments

**DOI:** 10.3390/md21090479

**Published:** 2023-08-30

**Authors:** Jongkyu Kim, Namju Lee, Yoon-Seok Chun, Sang-Hoon Lee, Sae-Kwang Ku

**Affiliations:** 1AriBnC Co., Ltd., Yongin 16914, Republic of Korea; jkkim@aribnc.com (J.K.); ceochun@aribnc.com (Y.-S.C.); 2Department of Veterinary Surgery, College of Veterinary Medicine, Kyungpook National University, Daegu 41566, Republic of Korea; form23h@knu.ac.kr; 3Department of Anatomy and Histology, College of Korean Medicine, Daegu Haany University, Gyeongsan 38610, Republic of Korea

**Keywords:** krill oil, ultraviolet, skin photoaging, marine-derived ingredients

## Abstract

Krill oil (KO) shows promise as a natural marine-derived ingredient for improving skin health. This study investigated its antioxidant, anti-inflammatory, anti-wrinkle, and moisturizing effects on skin cells and UVB-induced skin photoaging in hairless mice. In vitro assays on HDF, HaCaT, and B16/F10 cells, as well as in vivo experiments on 60 hairless mice were conducted. A cell viability assay, diphenyl-1-picryhydrazyl (DPPH) radical scavenging activity test, elastase inhibition assay, procollagen content test, MMP-1 inhibition test, and hyaluronan production assay were used to experiment on in vitro cell models. Mice received oral KO administration (100, 200, or 400 mg/kg) once a day for 15 weeks and UVB radiation three times a week. L-Ascorbic acid (L-AA) was orally administered at 100 mg/kg once daily for 15 weeks, starting from the initial ultraviolet B (UVB) exposures. L-AA administration followed each UVB session (0.18 J/cm^2^) after one hour. In vitro, KO significantly countered UVB-induced oxidative stress, reduced wrinkles, and prevented skin water loss by enhancing collagen and hyaluronic synthesis. In vivo, all KO dosages showed dose-dependent inhibition of oxidative stress-induced inflammatory photoaging-related skin changes. Skin mRNA expressions for hyaluronan synthesis and collagen synthesis genes also increased dose-dependently after KO treatment. Histopathological analysis confirmed that krill oil (KO) ameliorated the damage caused by UVB-irradiated skin tissues. The results imply that KO could potentially act as a positive measure in diminishing UVB-triggered skin photoaging and address various skin issues like wrinkles and moisturization when taken as a dietary supplement.

## 1. Introduction

The human skin, consisting of three layers (epidermis, dermis, and subcutaneous) [1], serves as a vital barrier against environmental stressors like ultraviolet (UV) rays, physical trauma, and microorganisms. Direct exposure to UV radiation from sunlight can lead to acute effects such as DNA damage, suppression of DNA synthesis, cell death, and erythema. Additionally, it can also result in chronic effects like photoaging and epidermal cancer [2]. Photoaging, induced by repeated UV exposure, results in histological alterations, collagen fiber damage, and uneven pigmentation, leading to wrinkled and coarse skin [3]. Ultraviolet B (UVB), with a wavelength range of 280 to 315 nanometers, exhibits both beneficial and harmful effects on the skin. It can cause sunburns, premature aging, and an increased risk of skin cancer, among other characteristics [4]. UVB penetrates the epidermis, leading to DNA damage and mutations over time, contributing to skin cancer development [5].

UV rays induce intracellular reactive oxygen species (ROS), causing extracellular matrix (ECM) components imbalance, inflammation, and immunosuppression, leading to skin photoaging [6]. Supplementing the diet with vitamins, minerals, or essential fatty acids that are lacking is thought to enhance the condition of the skin, influenced by factors like UV irradiation, free radicals, toxic substances, allergic compounds, and other environmental elements, as well as genetic predisposition, immune and hormone status, and stress, which can all cause damage to the skin. This is because the health and attractiveness of the skin are dependent on proper nutrition [7]. In addition, there is established knowledge regarding essential fatty acids like alpha-linolenic acid (ALA) and gamma-linolenic acid, which belong to the omega-3 and omega-6 classifications. The important roles of docosahexaenoic acid (DHA) and eicosapentaenoic acid (EPA), both derived from ALA, are widely acknowledged in influencing the structure and operation of the skin [8].

Krill oil (KO) is derived from krill (*Euphausia superba*), which is a small marine crustacean that feeds on marine algae. Unlike fish oil, which is primarily composed of triglycerides, KO consists of both phospholipids (30–65%) and triglycerides [9]. The main phospholipid found in KO is phosphatidylcholine (PC), and approximately 40% of the total fatty acids bound to PC are known to be EPA and DHA [10]. Therefore, it has been reported that EPA and DHA from KO exhibit higher bioavailability compared to other forms of n-3 PUFAs (such as ethyl-ester and re-esterified omega-3). Although KO has a smaller daily dose of n-3 PUFAs compared to fish oil, it demonstrates higher absorption rates. Hence, KO and fish oil can be considered comparable dietary sources of n-3 PUFAs, even though the EPA and DHA dose in krill oil is 62.8% of that in fish oil. [9]. Recently, KO has been studied not only for its role as a dietary supplement in improving human health, but also for its potential in the prevention and/or treatment of skin-related conditions [11]. It is suggested that KO may have beneficial effects on skin health and could be explored as a potential intervention for various skin concerns. Previous studies focused on various skin conditions have utilized the keratinocyte line (HaCaT) as a foundational element. To establish a more comprehensive range of evidence regarding the broader skin protective effects of KO, it is considered essential to conduct research at different cellular levels. Thus, we propose that conducting research involving HaCaT Cells (Human Keratinocytes) for skin response assessment, HDF Cells (Human Dermal Fibroblasts) for wound healing and aging studies, and B16/F10 Cells (Murine Melanoma Cells) for melanoma research would be advantageous.

While KO has been studied for its antioxidant and anti-inflammatory properties, more research is needed to understand its specific effects on skin under environmental stress and its potential to mitigate their impact. The goal of this study is to examine how the combination of KO and vitamin C (L-ascorbic acid, L-AA)—with specific reference to L-AA—influences the beneficial aspects of skin health. Due to the existing knowledge that L-AA enhances the skin barrier, reduces wrinkles, offers UV protection, inhibits melanogenesis, and provides moisturizing and antioxidant benefits, this study aims to investigate the efficacy of KO with L-AA as a reference in order to further explore its effects [8]. This study supports KO’s (Superba^TM^ Boost) positive impact on skin health, including wrinkle improvement, skin moisturization, inflammation reduction, and antioxidant activities. Using a UVB-induced skin photoaging mouse model and in vitro experiments, the study investigates skin photoaging and evaluates KO’s potential as a skin protection agent. 

## 2. Results

### 2.1. In Vitro Evidence Anti-Aging

#### 2.1.1. Cytotoxicity of KO in HDF, HaCaT, and B16/F10 Cells

To assess the safety within the natural source and evaluate efficacy within a secure range, quantification of cell viability is examined. To assess the cytotoxic effects of KO and determine suitable concentrations for subsequent cell-based assays, MTT assays were conducted on HDF, HaCaT, and B16/F10 cell lines. Across seven different concentrations (0.25, 0.5, 1, 1.5, 2, 4, and 8 mg/mL) of KO treatment, no significant alterations in cell viabilities were observed compared to the vehicle control after 48 h of incubation. Consequently, the IC50 values of KO for HDF, HaCaT, and B16/F10 cell viabilities were found to be > 8 mg/mL, at least within the conditions of the current analysis (Figure 1). For HDF cells, viabilities changed by −1.00 ± 9.36%, −1.17 ± 6.88%, 5.33 ± 8.29%, 0.17 ± 5.85%, 0.00 ± 6.45%, 1.50 ± 4.28%, and −1.33 ± 7.53% for KO treatments of 0.25, 0.5, 1, 1.5, 2, 4, and 8 mg/mL, respectively, compared to the vehicle controls. In the case of HaCaT cells, viabilities changed by −1.83 ± 8.16%, 1.33 ± 11.71%, 2.17 ± 6.27%, −0.50 ± 6.25%, 3.67 ± 5.85%, 1.33 ± 9.09%, and −1.67±7.09% for KO treatments of 0.25, 0.5, 1, 1.5, 2, 4, and 8 mg/mL, respectively, compared to the vehicle controls. For B16/F10 cells, viabilities changed by 3.33 ± 17.19%, −1.67 ± 8.36%, 4.83 ± 15.24%, 0.50 ± 9.99%, −0.83 ± 15.96%, 3.33 ± 16.43%, and −1.83 ± 8.01% for KO treatments of 0.25, 0.5, 1, 1.5, 2, 4, and 8 mg/mL, respectively, compared to the vehicle controls (Figure 1).

#### 2.1.2. Free Radical Scavenging Activity of KO

The evaluation of DPPH radical scavenging activities for L-AA (1 mg/mL) and KO (0.25, 0.5, 1, 1.5, 2, 4, and 8 mg/mL) showed that L-AA exhibited a scavenging activity of 82.00 ± 15.28%, and KO demonstrated significant increases in DPPH radical scavenging activity compared to the control: 27.17 ± 11.14, 49.50 ± 8.14, 59.67 ± 9.20, 63.67 ± 14.80, 75.33 ± 12.23, 81.17 ± 11.65, and 88.33 ± 8.80% (*p* < 0.01) (Figure 2).

#### 2.1.3. Anti-Wrinkle Benefits of KO

Phosphoramidon disodium salt (PP, 10 μM) showed a significant increase in elastase inhibitory activity by 38.17 ± 8.68% compared to the control (*p* < 0.01). Additionally, KO demonstrated a concentration-dependent increase in elastase inhibitory activity (17.17 ± 10.48, 31.50 ± 9.73, 54.50 ± 11.73, 64.50 ± 13.03, 77.50 ± 9.65, and 85.67 ± 7.45%) compared to the control (*p* < 0.01) (Figure 3a). At TGF-β1 10 ng/mL, there was a significant increase of 265.33 ± 31.33% in HDF cell procollagen synthesis compared to the control (*p* < 0.01). Moreover, KO showed a concentration-dependent (KO 0.25 to 8 mg/mL) increase in HDF cell procollagen synthesis (138.17 ± 18.40, 170.00 ± 30.10, 209.33 ± 26.52, 243.00 ± 19.59, 259.33 ± 24.74, 263.83 ± 25.76, and 272.00 ± 27.82%) compared to the control (*p* < 0.01) (Figure 3b). Under UVB irradiation (50 mJ/cm^2^), a significant increase in MMP-1 activity by 165.83 ± 10.48% was observed compared to non-irradiated UVB control cells (100.83 ± 4.92%) (*p* < 0.01). However, when treated with RA 1μM, a significant decrease in MMP-1 activity (128.67 ± 18.78%) was observed compared to UVB radiation (*p* < 0.01). Similarly, treatment with KO from 0.5 mg/mL to the highest concentration of 8 mg/mL resulted in a significant decrease in MMP-1 activity compared to UVB radiation (146.83 ± 9.04, 135.33 ± 10.41, 124.33 ± 7.97, 119.17 ± 13.57, 115.83 ± 15.93, 110.67 ± 11.84, and 109.00 ± 1 2.62%) (*p* < 0.01) (Figure 3c).

#### 2.1.4. Moisturizing Benefits of KO

In HaCaT cells, the evaluation of KO’s moisturizing activity through hyaluronan synthesis showed that RA at 1 μM resulted in a significant increase in hyaluronan synthesis (209.17 ± 17.44%) compared to the control (100.67 ± 5.35%) (*p* < 0.01). Similarly, KO demonstrated a concentration-dependent (KO 0.25, 0.5, 1, 1.5, 2, 4, and 8 mg/mL) increase in hyaluronan synthesis (153.83 ± 16.83, 207.00 ± 18.49, 211.17 ± 34.00, 233.33 ± 26.23, 255.67 ± 16.16, 263.17 ± 31.18, and 271.00 ± 29.85%) compared to the control (*p* < 0.01) (Figure 4).

### 2.2. In Vivo Evidence of Anti-Photoaging

#### 2.2.1. Effects of KO on Body Weight Changes

The intact control group’s average weight decreased from 24.93 ± 1.23 g to 21.77 ± 1.21 g on Day 0, and then increased to 32.82 ± 2.21 g on Day 104 (Figure 5). The UVB-exposed hairless mice showed no significant weight difference compared to non-UVB-exposed controls during the 105-day treatment. KO and L-AA groups also did not exhibit significant weight changes compared to intact controls or the UVB group. Additionally, no significant weight or mass index changes were observed in the UVB control group over the 105 days. Similarly, experimental substance-treated groups did not show significant changes. The body mass index showed a −4.51% change in the UVB control compared to normal medium controls. For L-AA 100 mg/kg, KO 400 mg/kg, 200 mg/kg, and 100 mg/kg groups, the changes were 2.29%, 6.15%, 2.86%, and 3.15%, respectively, compared to the UVB control.

#### 2.2.2. Effects of KO on UVB-Induced Skin Wrinkle Formation and Skin Moisturization

The efficacy of KO in examining wrinkle formation on UVB-irradiated skin was conducted, and the length and depth of skin wrinkles were evaluated using replicas of the dorsal back skin. Exposure to UVB radiation clearly induced skin wrinkles, but their severity was alleviated by oral administration of either KO or L-AA (Figure 6a). After exposure to UVB, the mean skin length (mm) and depth (µm) significantly increased (0.69 ± 0.07 mm, 99.16 ± 17.37 µm) compared to the intact control (0.29 ± 0.05 mm, 31.16 ± 10.45 µm) (*p* < 0.01). However, both KO and L-AA significantly suppressed the formation of skin wrinkles in terms of length and depth (*p* < 0.01) (Figure 6b, c). Skin water content significantly decreased by 13.16 ± 2.23% after UVB irradiation compared to the intact control group (37.56 ± 5.47%) (*p* < 0.01). However, both KO (*p* < 0.05) and L-AA (*p* < 0.01) treatments significantly increased skin water content (Figure 6d). Key molecules for skin wrinkle formation and skin moisture maintenance—skin COL1 contents and hyaluronic acid contents—were significantly increased by both L-AA and KO treatments compared to UVB irradiation alone (*p* < 0.01) (Figure 6e,f). Moreover, the mRNA expression of hyaluronic acid and COL1-related genes *COL1A1*, *COL1A2*, *Has1*, *Has2*, and *Has3* showed a significant increase compared to UVB irradiation alone (*p* < 0.01) (Figure 6g,h). Transforming growth factor-beta (TGF-β) stimulates collagen formation through the regulation of various cellular functions, with TGF-β1 particularly promoting the synthesis of hyaluronan via *Has1* and *Has2 mRNA* expression. Levels of *TGF-β1 mRNA* were significantly decreased in UVB irradiation (0.17 ± 0.03) compared to the intact control group (1.00 ± 0.05) (*p* < 0.01). However, a significant dose-dependent increase in *TGF-β1 mRNA* expression was detected in the oral administration of KO (100, 200, and 400 mg/kg) compared to UVB irradiation (*p* < 0.01) (Figure 6i). The mRNA expression of UVB-induced *MMP1, MMP9*, and *MMP13* was significantly reduced by both L-AA and KO treatments (*p* < 0.01) (Figure 6j). There was no significant difference in the treatment effects of UVB-induced skin wrinkle formation and skin moisturization between KO and L-AA treatments with an equal oral dosage. 

#### 2.2.3. Effects of KO on UVB-Induced Skin Inflammation

Continuous exposure to UVB irradiation can lead to skin inflammation, causing symptoms such as erythema, swelling, edema, and itching. The evaluation of KO’s effect on skin edema was performed by measuring the weight of 6 mm diameter skin samples. The increased weight of skin samples due to UVB irradiation (0.105 ± 0.017 g) was significantly reduced by both L-AA (0.064 ± 0.011 g) and KO (KO100, 0.075 ± 0.010 g; KO200, 0.063 ± 0.007 g; KO400, 0.048 ± 0.009 g) treatments (*p* < 0.01) (Figure 7a). Moreover, UVB irradiation (14.10 ± 3.07 neutrophils × 105/mg protein) leads to a significant increase in MPO activity compared to the intact control (1.55 ± 0.40 neutrophils × 105/mg protein). However, both L-AA (5.99 ± 1.81 neutrophils × 105/mg protein) and KO treatments (KO100: 7.73 ± 1.35 neutrophils × 105/mg protein; KO200: 5.98 ± 1.64 neutrophils × 105/mg protein; KO400: 3.81 ± 1.51 neutrophils × 105/mg protein) significantly reduce this activity (*p* < 0.01) in comparison to the intact control (1.55 ± 0.40 neutrophils × 105/mg protein) (Figure 7b). UVB irradiation resulted in a notable increase in IL-1β levels (79.04 ± 11.96 pg/100 mg of protein) when compared to the intact control (22.53 ± 5.38 pg/100 mg of protein). However, these elevated levels were significantly suppressed by both L-AA (43.33 ± 10.25 pg/100 mg of protein) and KO treatments (KO100: 50.23 ± 11.52 pg/100 mg of protein; KO200: 42.32 ± 10.30 pg/100 mg of protein; KO400: 35.25 ± 10.63 pg/100 mg of protein) (*p* < 0.01) in comparison to UVB irradiation. Conversely, UVB irradiation led to a notable reduction in IL-10 levels (123.60 ± 26.90 pg/100 mg of protein) when compared to the intact control (356.10 ± 70.68 pg/100 mg of protein). In contrast, both L-AA (241.00 ± 62.36 pg/100 mg of protein) and KO treatments (KO100: 201.90 ± 57.04 pg/100 mg of protein; KO200: 244.70 ± 42.06 pg/100 mg of protein; KO400: 279.60 ± 30.34 pg/100 mg of protein) significantly increased these levels (*p* < 0.05) when compared to UVB irradiation (Figure 7c). *p38 MAPK* and *AKT mRNA* were significantly inhibited following oral administration of all three doses of KO compared to UVB irradiation (*p* < 0.05) (Figure 7d,e). There was no significant difference in the treatment effects of UVB-induced skin inflammation between KO and L-AA treatments with an equal oral dosage.

#### 2.2.4. Effects of KO on UVB-Induced Oxidative Stress

The evaluation of krill oil’s antioxidant effects on skin tissue involved investigating alterations in the levels of glutathione (GSH), malondialdehyde (MDA), and superoxide anions. UVB irradiation (0.27 ± 0.09 uM/mg of protein) led to a significant decrease in GSH content compared to the intact control (1.73 ± 0.25 μM/mg of protein), but administration of L-AA (0.60 ± 0.11 μM/mg of protein) and KO (KO100, 0.48 ± 0.06 μM/mg of protein; KO200, 0.61 ± 0.12 μM/mg of protein; KO400, 0.83 ± 0.15 μM/mg of protein) significantly increased GSH levels (*p* < 0.05) (Figure 8a). Additionally, the increased MDA and superoxide anion levels caused by UVB irradiation were significantly suppressed by the administration of L-AA and KO (*p* < 0.01) (Figure 8b,c). *GSH reductase mRNA* expression, which was significantly reduced by UVB irradiation (0.35 ± 0.10) compared to the intact control (1.00 ± 0.06), was significantly increased by the administration of L-AA (0.66 ± 0.11) and KO (KO100, 0.59 ± 0.09; KO200, 0.67 ± 0.11; KO400, 0.83 ± 0.12) (*p* < 0.01) (Figure 8d). Conversely, UVB irradiation (3.68 ± 0.70) markedly elevated the mRNA expression of NOX2, an enzyme associated with NADPH oxidase-derived ROS generation, in comparison to the intact control (1.00 ± 0.05). However, both L-AA (2.08 ± 0.48) and KO treatments (KO100: 2.62 ± 0.39; KO200: 2.03 ± 0.32; KO400: 1.59 ± 0.27) significantly reduced this expression (*p* < 0.05) (Figure 8e). There was no significant difference in the treatment effects of UVB-induced skin oxidative stress between KO and L-AA treatments with an equal oral dosage. 

#### 2.2.5. Effects of KO on UVB-Induced Histopathological Changes in Skin Tissue

Histopathological analysis and Masson’s trichrome staining revealed thickened epithelial tissue and abnormal collagen deposition in the dorsal back skin tissue due to UVB irradiation. However, the histopathological changes were significantly mitigated by L-AA and KO treatments (Figure 9a and Table 1). Additionally, KO treatments improved UVB-induced immunolabeled cells for oxidative stress markers (NT and 4-HNE), apoptosis markers (cleaved caspase 3 and cleaved PARP) in the epidermis, and immunoreactive cells for MMP9 in the dermis (Figure 9b and Table 2). However, there was no significant difference in the treatment effects of UVB-induced histopathological changes in skin tissue between KO and L-AA treatments with an equal oral dosage.

## 3. Discussion

Skin aging can be categorized into intrinsic aging, driven by hormonal changes and cellular aging, and extrinsic aging, caused by external factors like UV exposure, air pollution, and smoking [12]. As the demand for anti-skin-aging solutions increases in the market, research and development of natural product-derived ingredients have advanced cutaneous science in skin beauty and health-related industries. Nutricosmetic products, including UV protectors, anti-wrinkle treatments, and moisturizers, are introduced to address these concerns [8]. However, the long-term consumption of cost-effective and functional products raises ongoing concerns about potential risks of adverse events, harmful chemicals, and toxins [13]. 

Krill oil (KO) is gaining attention for its high bioavailability of n-3 polyunsaturated fatty acids (PUFAs) like EPA and DHA in phospholipid form. Despite being more expensive than fish oil, the superior bioavailability of EPA/DHA has sparked interest. Some clinical studies [9,14,15] have reported minor adverse events, such as rashes, headaches, taste changes, diarrhea, and a decreased appetite. However, KO is generally recognized as safe (GRAS) by the American Food and Drug Administration and has received Novel Food status from the European Union, confirming its safety profile [10]. 

In our previous studies, KO has been recognized as a marine-derived natural substance with significant activation of nuclear factor E2-related factor 2 (Nrf2) transferase and potent antioxidant properties, making it a promising raw material for health functional foods with natural antioxidant benefits [16,17]. Although previous research on human immortalized keratinocyte lines and NC/Nga mice has suggested the potential benefits of KO in terms of antioxidants and anti-inflammatory effects [11,18], there is a lack of direct and detailed research on KO’s role in improving skin wrinkles and moisturization. Hence, this study aims to scientifically evaluate the anti-skin-aging effects of KO.

Skin aging is linked to the activation of matrix metalloproteinases (MMPs) triggered by inflammatory cytokines in skin tissue. Frequent exposure to UV radiation accelerates skin aging by causing DNA breakdown, ROS generation, and DNA damage [19]. Nutricosmetic products, known for their antioxidant and anti-inflammatory functions, are anticipated to prevent or improve skin aging. Enzymatic antioxidants have been shown to reduce UV-induced oxidative stress in skin tissue, suppressing inflammation and inhibiting apoptosis of skin cells [7,19]. KO may not be an enzymatic antioxidant; our data demonstrated its inhibition of apoptosis in UVB-exposed skin tissue of mice, suggesting KO’s potential in suppressing skin inflammation from UV radiation. Moreover, KO’s inhibitory effects may extend to the apoptosis pathway and cell cycle arrest caused by UV-induced DNA damage [20], specifically cyclobutane pyrimidine dimers (CPD) and pyrimidine 6-4 pyrimidine photoproducts (6-4PP). Further research on KO’s effects on CPD and 6-4PP as well as its ability to modulate CPD-photolyases and 6-4PP-photolyase repairing mechanisms would provide a clearer understanding of KO’s protection against UV-induced DNA damage and apoptosis. These investigations will enhance our understanding of KO’s potential as a skin-protective agent.

Previous studies have reported KO’s antioxidant and anti-inflammatory effects on human immortalized keratinocyte lines [11]. KO’s potential to regulate ECM proteins and protect the skin through its antioxidant activity and anti-inflammatory effects was supported by its ability to suppress skin inflammation in NC/Nga mice using a phospholipid-enriched alkyl phospholipid from krill [18]. As a non-enzymatic antioxidant, KO builds an antioxidant defense network, protecting cells and tissues from ROS and benefiting skin health. In this study, we observed KO’s concentration-dependent free radical scavenging activity. Furthermore, KO administration improved the UVB-induced decrease in GSH content by upregulating *GSH reductase mRNA* expression. KO also inhibited UVB-induced lipid peroxidation and superoxide anion production through the transcriptional regulation of *NOX2*. The observed results, confirmed through immunohistochemical analysis using NT and 4-HNE staining, suggest that KO’s antioxidant activity plays a significant role. Additionally, increased ROS due to UV exposure can activate the MAPK (mitogen-activated protein kinase) signaling pathway, leading to the activation of AP-1 (activated protein-1) and subsequently promoting the expression of MMPs (matrix metalloproteinases), which can strongly contribute to the breakdown of ECM proteins like collagen and elastin [21,22,23,24]. Indeed, the increased MMPs due to UV radiation can promote the degradation of ECM proteins and ultimately lead to the formation of skin wrinkles and photoaging [24]. While our study did not directly investigate the UV-induced MAPK pathway and AP-1 activation, we observed that KO’s antioxidant activity effectively suppressed the mRNA expression of *MMP-1, MMP-9*, and *MMP-13*. This inhibition contributed to the regulation of ECM proteins and directly prevented the formation of skin wrinkles, evident by the mean length and depth of wrinkles.

Human skin contains 28 different types of MMPs, including collagenases (MMP-1 and MMP-13) and a gelatinase (MMP-9), which increase with UV exposure [25]. MMP-1 and MMP-13 not only promote ECM collagen degradation but also reduce collagen density in the dermal layer [23,26]. Our study demonstrated that KO reduced UVB-induced MMP-1 activity in HDF cells, indicating its potential to inhibit ECM collagen degradation. Moreover, KO suppressed the upregulation of *MMP-1*, *MMP-9*, and *MMP-13 gene* expression in skin tissue induced by UVB, leading to improved skin COL1 levels and *COL1A1/2 mRNA* expression. This suggests that KO may inhibit *MMP mRNA* expression, likely through its effect on local inflammatory and neutrophil responses to UVB [27]. Such MMP suppression could help maintain skin collagen levels and prevent excessive collagen degradation linked to photoaging. Additionally, our findings show that UV radiation induces an inflammatory response in the skin, with increased secretion of pro-inflammatory cytokine IL-1 and reduced expression of anti-inflammatory cytokine IL-10. KO administration appears to regulate this inflammatory state induced by UV, balancing pro- and anti-inflammatory cytokines, and contributing to both its anti-inflammatory effects and its potential skin health benefits.

In response to UVB-induced skin injury, polymorphneutrophils (PMNs) and neutrophils are recruited to the injured tissues through the action of oxygen metabolites [28]. MPO, released from PMNs, is a cytotoxic enzyme that activates inflammation [29]. The reduction of neutrophils infiltrating into the skin tissue can be confirmed by assessing MPO activity [30]. Our study revealed that KO alleviates the UVB-induced inflammatory state in the skin and directly inhibits MPO activity, leading to reduced neutrophil recruitment to inflammatory sites [31]. These findings suggest that KO administration modulates the inflammatory response induced by UVB exposure by decreasing MPO activity and subsequently limiting neutrophil infiltration into the skin.

Skin aging often leads to a reduction in hyaluronic acid, a vital component responsible for retaining water in the skin [32]. Fatty acids play a crucial role in maintaining skin hydration and barrier integrity [8], while PUFA deficiency can increase water loss through the skin barrier [33]. Our study found that UVB exposure and aging downregulated the genes responsible for hyaluronic acid synthesis (*HAS1*, *HAS2*, and *HAS3*) in the dermis [34]. However, oral administration of KO reversed this downregulation, resulting in increased hyaluronic acid content in the skin. These results suggest that KO enhances skin moisturization by promoting hyaluronic acid synthesis through the regulation of *HAS genes* in response to UVB-induced water loss.

This study was conducted with the objective of investigating the protective effects of KO against UVB-induced skin photoaging. The existing literature has only offered limited insights into the potential skin health advantages of KO, with two or fewer studies referencing it. Our investigation demonstrated that oral administration of KO notably mitigated UVB-induced wrinkles, skin water loss, collagen degradation, and skin edema, comparable to L-AA (100 mg/kg) at the same dosage. These findings indicate the potential of KO as a functional product for preventing UVB-induced skin photoaging and enhancing skin moisturization. However, further clinical studies are necessary to comprehensively elucidate the diverse range of benefits provided by KO for skin health.

## 4. Materials and Methods

### 4.1. In Vitro

#### 4.1.1. Preparation of KO

The commercial product of Antarctic Krill Oil (KO; Superba^TM^ Boost) was produced by Arker Biomarine (Houston, TX, USA) and was supplied by SC Science (Goyang, Republic of Korea) for the study. KO extracted *Euphausia superba* through steam heating followed by ethanol extraction. The solid particles were separated through filtration and then refined by adding ion exchange resin and NaOH. The refined KO was further processed at 60 °C and under 360 mmHg vapor pressure for 1 h to remove ethanol. After centrifugation, a secondary evaporation and filtration were conducted to produce the final product, KO. The KO composition included 51.2% (wt/wt) phospholipids, comprising 44.9% phosphatidylcholine, 3.6% 1-palmitoyl-2-hydroxy-glycero-3-phosphocholine, 2.1% phosphatidylethanolamine, and 0.6% N-Acyl-phosphatidylethanolamine [17]. Additionally, EPA and DHA contents were analyzed in the KO. Methanol-treated KO was subjected to sodium hydroxide and boron trifluoride esterification, followed by dissolution in isooctane for analysis using gas chromatography (GC). A gas chromatograph (Agilent, USA) equipped with an SP^®^-2560 capillary GC column (100 m × 0.25 mm, 0.20 μm) and a flame ionization detector (GC/FID) was used. Helium was employed as the carrier gas at a flow rate of 0.75 mL/min with a split ratio of 200:1. The injector and detector (FID) temperatures were set at 225 °C and 285 °C, respectively. The column temperature was maintained at 100 °C for 4 min, followed by an increase to 240 °C at a rate of 3 °C/min. Quantification was achieved by calculating the peak areas of each fatty acid obtained from the test and standard solutions, along with the peak area of internal standard substances. The EPA and DHA content in the KO was determined to be 296 mg/g (Appendix A). The KO was stored at −20 °C and utilized consistently for in vitro and in vivo studies.

L-Ascorbic acid (Vitamin C; L-AA), retinoic acid (RA), and phosphoramidon disodium salt (PP) were purchased from Sigma-Aldrich (St. Louis, MO, USA), while TGF-β1 was obtained from R&D Systems (Minneapolis, MN, USA). The treatment concentrations for KO in the study were 0.25, 0.5, 1, 1.5, 2, 4, and 8 mg/mL, while the treatment concentrations for the reference compounds, namely L-AA (1 mg/mL), RA (1 μM), PP (10 μM), and TGF-β1 (10 ng/mL), were each as follows.

#### 4.1.2. Cell Cultures

Human keratinocyte (HaCaT) and murine melanoma cells (B16/F10) were procured from the American Type Culture Collection (ATCC, Manassas, VA, USA) and cultured following the manufacturer’s protocols. The cells were maintained in Dulbecco’s modified Eagle’s medium (DMEM; Sigma-Aldrich, St. Louis, MO, USA) supplemented with 10% fetal bovine serum (FBS; Lonza, Walkersville, MD, USA), 100 µg/mL streptomycin, and 100 U/mL penicillin (Sigma-Aldrich, St. Louis, MO, USA). For B16/F10 murine melanoma cells, 2 mM L-glutamine (Sigma-Aldrich, St. Louis, MO, USA) was additionally added to the culture medium. Human dermal fibroblasts (HDF) and neonatal cell lines were also obtained from ATCC (Manassas, VA, USA) and cultured in a fibroblast basal medium (FBM; PCS-201-030, ATCC, Manassas, VA, USA) supplemented with a FBM low serum kit (PCS-201-041; ATCC, Manassas, VA, USA). For treatments, a FBM serum-free kit (PCS-201-040; ATCC, Manassas, VA, USA) was used. All cells were cultured at 37 °C in a fully humidified atmosphere with 5% CO_2_ using a commercial CO_2_ incubator (Model 311, Thermo Forma, Marietta, OH, USA), and were passaged approximately every other day.

#### 4.1.3. Cell Viability Assay

HaCaT, HDF, and B16/F10 cells were seeded at 1 × 10^5^ cells/well in 96-well plates and exposed to various concentrations of KO (0.25, 0.5, 1, 1.5, 2, 4, and 8 mg/mL) for 48 h. Cell viability was assessed using an MTT reagent (Sigma-Aldrich, St. Louis, MO, USA) with a concentration of 2 mg/mL. The plates were then incubated in a CO_2_ incubator at 37 °C for 2 h. To measure the cell viability, the absorbance (optical density, OD) of the wells at 450 nm was recorded using a microplate reader (Sunrise, TECAN, Männedorf, Switzerland). The relative cell viability (%) was calculated as [(OD_s_/OD_c_) × 100], where OD_s_ represents the absorbance of the sample at 450 nm, and OD_c_ is the absorbance of the vehicle control at 450 nm. The results were expressed in terms of inhibitory concentration (IC)_50_, which indicates the concentration at which cell viability reaches 50% of the control.

#### 4.1.4. DPPH Radical Scavenging Assay

The KO’s free radical scavenging ability was assessed through the DPPH radical scavenging assay, following the method outlined by Blois [35]. A 0.2 mM solution of 2,2-diphenyl-1-picryhydrazyl (DPPH; Sigma-Aldrich, St. Louis, MO, USA) in methanol was promptly prepared. Samples were diluted using distilled water to reach final KO concentrations of 0.25, 0.5, 1, 1.5, 2, 4, and 8 mg/mL, or a final L-AA concentration of 1 mg/mL. The DPPH radical scavenging activity was measured at 517 nm with a UV/Vis spectrophotometer (Optizen Pop, Mecasys, Daejeon, Republic of Korea) after a 10-min incubation. The free radical scavenging activity was calculated using the formula: DPPH radical scavenging activity (%) = 100 − [(OD_s_/OD_c_) × 100], where OD_s_ represents the sample’s absorbance at 517 nm and ODc is the absorbance of the control treated with the vehicle at 517 nm. The results were expressed as IC_50_ values, indicating the concentration needed to decrease DPPH by 50%. A positive control of L-AA at 1 mg/mL was employed.

#### 4.1.5. Elastase Inhibition Assay

The elastase inhibition assay measured the release of p-nitroaniline due to proteolysis of N-succinyl-(Ala)3-p-nitroanilide by the human leucocyte elastase (Sigma-Aldrich, St. Louis, MO, USA). KO was tested at concentrations ranging from 0.25 to 8 mg/mL, while PP at 10 μM served as the standard. Elastase inhibitory activity was measured at 410 nm using a 96-well microplate reader, and the elastase inhibitory activity of each sample was calculated using equation as follows: Elastase inhibitory activity (%) = 100 − [(OD_s_/OD_c_) × 100], where OD_s_ is the absorbance of the experimental sample at 410 nm and OD_c_ is the absorbance of the vehicle-treated control at 410 nm. The results were reported as IC_50_, representing the concentration at which the percentage inhibition of elastase activity was 50%.

#### 4.1.6. Procollagen Synthesis Assay

HDF cells were cultured in 24-well plates (2 × 10^4^ cells/well) for 24 h. Subsequently, the medium was replaced with varying concentrations of KO (0.25, 0.5, 1, 1.5, 2, 4, and 8 mg/mL) or 10 ng/mL TGF-β1 mixed with a serum-free kit, and cells were cultured for another 24 h. Procollagen levels in the culture supernatant were measured using a Procollagen type I-c-peptide (PIP) ELISA kit (MK101, Takara Bio, Tokyo, Japan), normalized by total protein content. Relative procollagen synthesis (%) was calculated as [(procollagen contents in experimental sample/procollagen contents in control) × 100]. The results were presented as the 50% effective concentrations (EC_50_), representing the concentration at which HDF cell procollagen synthesis doubled.

#### 4.1.7. MMP-1 Activity

Using a fluorescence microplate reader (RF-5301PC; Shimadzu Corp., Tokyo, Japan) and adapting Losso et al.’s method [36], HDF cells were cultured (2 × 10^4^ cells/well) for 24 h. Following this, cells were exposed to KO (0.25, 0.5, 1, 1.5, 2, 4, and 8 mg/mL) or 1 μM RA, with or without 5 mJ/cm^2^ UVB irradiation for 2 min and cultured for an additional 24 h. Total MMP-1 protein levels in supernatants were quantified using the Human Total MMP-1 ELISA kit (DY901; R&D Systems, Minneapolis, MN, USA), normalized to total protein content. Relative MMP-1 expressions (%) were calculated as [(MMP-1 contents in UVB-exposed control or experimental sample/MMP-1 contents in unexposed control) × 100]. IC_50_ values were reported as concentrations where UVB-induced HDF cell MMP-1 expressions were 50% inhibited. UVB irradiation was performed using a UV irradiation Crosslinker system (CL-1000M, Analytik Jena, Upland, CA, USA).

#### 4.1.8. Hyaluronan Production Assay

HaCaT cells (4 × 10^4^ cells/well) were exposed to KO (0.25, 0.5, 1, 1.5, 2, 4, and 8 mg/mL) or 1 μM of RA for 24 h. Following this, the cells were trypsinized and counted for normalization purposes. The quantification of hyaluronan synthesis was conducted using the Hyaluronan ELISA kit (DY3614, R&D Systems, Minneapolis, MN, USA). The results were normalized based on the total protein content of the supernatant. The relative hyaluronan synthesis (%) was calculated as [(hyaluronan contents in the experimental sample/hyaluronan contents in the unexposed control) × 100]. The outcomes were presented as EC_50_ values, indicating the concentration at which the percentage increase in hyaluronan synthesis reached a two-fold level.

### 4.2. In Vivo

A total of 60 SPF/VAF Outbred SKH1-hr hairless female mice (OrientBio, Seungnam, Korea) were procured. The mice were housed in groups of five per polycarbonate cage within a controlled environment at a temperature of 20–25 °C and humidity of 50–55%. Following a 7-day acclimation period, mice with normal skin and stable body weight (average 24.91 ± 1.14 g, range: 22.60–27.00 g) were sorted into six groups, each consisting of 10 mice. The animal experiment was carried out with prior approval from the Institutional Animal Care and Use Committee of Daegu Haany University [Approval No. DHU2021-070, 18 August 2021].

#### 4.2.1. Preparation of Experimental Group

The mice were divided into a total of 6 groups, with 10 mice per group: (1) Intact control group (UVB unexposed control; administered sterile distilled water orally); (2) UVB control group (UVB exposure control; administered sterile distilled water orally); (3) L-AA group (UVB exposure and L-AA 100 mg/kg oral administration); (4) KO 100 group (UVB exposure and KO 100 mg/kg oral administration); (5) KO 200 group (UVB exposure and KO 200 mg/kg oral administration); and (6) KO 400 group (UVB exposure and KO 400 mg/kg oral administration).

#### 4.2.2. Skin Photoaging

Following the methods used in previous studies [37,38], a UV Crosslinker system (CL-1000M, Analytik Jena, Upland, CA, USA) emitting wavelengths of 254 nm, 312 nm, and 365 nm (with 312 nm as the main wavelength) was used to induce skin photoaging in the hairless mice. The mice were exposed to UVB at a dose of 0.18 J/cm^2^, three times per week, for 15 weeks. For the intact control group, the same environment stress was applied by leaving the UV Crosslinker system powered off under the same conditions for the same duration.

#### 4.2.3. Experimental Substances and Oral Administration

The experiment substances were prepared by dissolving KO at concentrations of 10, 20, and 40 mg/mL in sterile distilled water. They were administered orally using a metal gavage needle attached to a 1 mL syringe at doses of 10 mL/kg (equivalence to 100, 200, and 400 mg/kg) daily for 105 days, 1 h after UVB exposure. Taking into account the oral administration method of L-AA as investigated by Park et al. [13], L-AA was also dissolved in sterile distilled water at a concentration of 10 mg/mL and administered orally at a dose of 10 mL/kg (equivalence to 100 mg/kg) daily for 105 days, 1 h after UVB exposure. In the intact control and UVB control groups, only sterile distilled water was administered orally at the same volume and duration as the experimental substances to apply the same handling stress. The experimental substances were prepared at least once a week and stored in a refrigerator at 4 °C until use.

#### 4.2.4. Body Weight Measurement

The individual body weight of each mouse was measured once a week, starting from one day before the initial UVB exposure and administration of the test substance, up to the point of termination. This was accomplished using an automatic electronic balance (XB320M, Precisa Instrument, Zuerich, Switzerland). All animals underwent an overnight fasting period (approximately 18 h, with access to water) prior to the initial administration of the test substance and at the time of termination. The one-day fasting prior to the start of the main experiment was implemented to maintain consistent conditions among the groups, thereby minimizing potential variations resulting from food intake. This fasting approach aimed to ensure uniformity across the groups and prevent interference with drug absorption through dietary intake during drug administration initiation. Additionally, to mitigate these individual differences, the weight gained by each mouse over the course of the 105-day experimental period was calculated using the following formula: Body weight gain during 105-day experimental period = Body weight at termination – Body weight at initial test substance administration (1 h after UVB irradiation).

#### 4.2.5. Anti-Wrinkle and Moisturizing

##### Generation of Replicas and Image Analysis

Mouse dorsal skin replicas were produced using the Repliflo Cartridge Kit (CuDerm Corp., Dallas, TX, USA) under 2–3% isoflurane anesthesia (Hana Pharm. Co., Hwasung, Republic of Korea) via a rodent inhalation anesthesia setup (Surgivet, Waukesha, WI, USA) and a rodent ventilator (Model 687, Harvard Apparatus, Cambridge, UK). A pre-sacrifice photo of the dorsal skin, specifically the gluteal region, was taken with a digital camera (FinePix S700, Fujifilm, Tokyo, Japan). Replicas were placed on a horizontal stand, and wrinkle shadows were induced by illuminating them at a 40° angle with constant-intensity light. Monochrome images were captured by a CCD camera and analyzed using the Skin-Visiometer system (SV600, Courage & Khazaka, Cologne, Germany). The assessment of wrinkles utilized parameters like average length and depth, following established methods with modifications [36,38].

##### Skin Water Content Measurement

Twenty-four hours after the conclusion of the 105th consecutive oral administration of test substances, 6 mm-diameter skin samples were extracted from the dorsal back. Skin water contents (%) were then assessed using an automated moisture analyzer balance (MB23, Ohaus, Pine Brook, NJ, USA).

##### Determination of COL1 and Hyaluronan Contents in Skin Tissue

In this study, the analysis of COL1 contents followed Kim et al. and Kang et al.’s methods [39,40]. Dorsal back skin sections were isolated 24 h after the 105th substance administration. These samples were homogenized with a bead beater (Taco ^TM^Pre, Gene Research Biotechnology Corp., Taichung, Taiwan), an ultrasonic cell disruptor (KS-750, Madell Technology Corp., Ontario, CA, USA), and a radioimmunoprecipitation assay (RIPA) buffer (Sigma-Aldrich, St. Louis, MO, USA). Supernatants were separated by centrifugation at 15,000 rpm under 4 °C using a cryocentrifuge (Labocene 1236 MGR, Gyrozen, Daejeon, Republic of Korea) and stored at −150 °C in an ultra-deep freezer (MDF-1156, Sanyo, Tokyo, Japan) for future analysis. The quantification of pro-COL1 was carried out using a pro-COL I C peptide assay kit (MK101, Takara Bio, Tokyo, Japan) following the manufacturer’s guidelines. The readings were taken at 450 nm using a microplate reader (Sunrise, Tecan; Männedorf, Switzerland).

For hyaluronan assessment, established methods were employed [41,42]. Dorsal back skin samples were defatted with acetone, dried, weighed, and boiled for 20 min in a 50 mM Tris/HCl (pH 7.8) buffer. They were then subjected to proteolytic digestion with 1% *w*/*v* actinase E (Sigma-Aldrich, St. Louis, MO, USA) for a week at 40 °C. Subsequent deproteinization was carried out by adding 10% *w*/*v* trichloroacetic acid (Sigma-Aldrich, St. Louis, MO, USA) followed by centrifugation at 3000 rpm at 4 °C for 20 min. The resulting supernatants were neutralized with 10 N NaOH. Hyaluronan levels were measured using the mouse hyaluronic acid enzyme-linked immunosorbent assay (ELISA) kit (MBS161603, Mybiosource, San Diego, CA, USA), with readings at 450 nm on a microplate reader in ng/mL terms.

#### 4.2.6. Antioxidant and Anti-Inflammatory

##### Glutathione (GSH) Assay

Cutaneous GSH levels were assessed using a fluorescence assay as previously outlined [31]. Initially, the skin (1:3, *w*/*w* dilution) was homogenized in 100 mM NaH_2_PO_4_ (pH 8.0; Sigma-Aldrich, St. Louis, MO, USA) containing 5 mM EDTA. Subsequently, the homogenates underwent treatment with 30% trichloroacetic acid (Sigma-Aldrich, St. Louis, MO, USA) and were subjected to two rounds of centrifugation (at 1940× *g* for 6 min and at 485× *g* for 10 min). The fluorescence of the resultant supernatant was measured using a fluorescence spectrophotometer (RF-5301PC; Shimadzu Corp., Tokyo, Japan). In total, 100 μL of the supernatant was mixed with 1 mL of buffer 1 and 100 μL of o-phthalaldehyde (1 mg/mL in methanol; Sigma-Aldrich, St. Louis, MO, USA). The fluorescence intensity was measured after 15 min (k*_exc_* = 350 nm; k*_em_* = 420 nm). A standard curve was established using different concentrations of GSH (ranging from 0.0 to 75.0 μM). Protein levels in the skin homogenates were quantified following the method of Lowry et al. [43]. The outcomes were expressed as μM of GSH per milligram of protein.

##### Lipid Peroxidation Assay

To begin, the protein content of the homogenate (10 mg/mL in 1.15% KCl) was assessed using the Lowry method [43]. To evaluate lipid peroxidation, the measurement of thiobarbituric acid reactive substances (TBARS) was employed as previously elucidated [44]. In this procedure, trichloroacetic acid (10%; Sigma-Aldrich, St. Louis, MO, USA) was added to the homogenate to precipitate proteins. Following this step, the mixture underwent centrifugation (3 min, 1000× *g*). The protein-free sample was then extracted, and thiobarbituric acid (0.67%) was introduced. This mixture was subjected to a water bath at 100 °C for 15 min. The intermediate product of lipoperoxidation, MDA, was quantified by measuring the difference between absorbances at 535 and 572 nm on a microplate spectrophotometer reader. The findings were reported as nM/mg of protein [45].

##### Superoxide Anion Production

The measurement of superoxide anion production in skin tissue homogenates (10 mg/mL in 1.15% KCl) was conducted through the nitroblue tetrazolium (NBT) assay [46]. In brief, 50 μL of the homogenate was incubated with 100 μL of NBT (1 mg/mL; Sigma-Aldrich, St. Louis, MO, USA) in 96-well plates at 37 °C for 1 h. Subsequently, the supernatant was cautiously removed, and the reduced formazan was dissolved by adding 120 μL of 2M KOH and 140 μL of DMSO. The reduction of NBT was measured at 600 nm using a microplate reader. The protein content was employed for the normalization of data.

##### Determination of IL-1β and IL-10 in Skin Tissues

On the 105th day of UVB irradiation, dorsal back skin tissue samples were obtained from the region around the gluteal area. Skin tissue homogenates were subjected to the measurement of IL-1β and IL-10 contents using mouse IL-1β (ab100705; Abcam, Cambridge, UK) and IL-10 (ab108870; Abcam, Cambridge, UK) enzyme-linked immunosorbent assay (ELISA) kits, following the manufacturer’s instructions. Optical density readings were taken at 450 nm using a microplate reader.

##### Edema Evaluation

The impact of the test substances on UVB-induced skin edema was evaluated by observing an increase in the weight of the dorsal skin. Following the continuous oral administration of the test articles for 15 weeks, the dorsal skin was removed. A consistent area (6 mm diameter) was then demarcated using a punch, and the weight of this standardized area was measured following established methodologies [13,47]. The outcome was determined by comparing the skin weight across different groups and was expressed in grams of skin (g/6-mm diameter of dorsal skin).

##### Myeloperoxidase (MPO) Activity

UVB-induced leukocyte migration to the skin was assessed using the MPO kinetic-colorimetric assay as per established protocols [31,48]. Skin samples were collected in 400 μL of a 50 mM K_2_HPO_4_ buffer (pH 6.0; Sigma-Aldrich, St. Louis, MO, USA) containing 0.5% hexadecyltrimethylammonium bromide (Gibco, Carlsbad, CA, USA), homogenized using a bead beater and ultrasonic disruptor, then centrifuged. The resulting supernatant was stored at −150 °C. For the assay, 30 μL of the sample was mixed with 200 μL of 0.05 M K_2_HPO_4_ buffer (pH 6.0) containing 0.167 mg/mL o-dianisidine dihydrochloride (Sigma-Aldrich, St. Louis, MO, USA) and 0.05% hydrogen peroxide. Absorbance was measured at 450 nm after 5 min using a UV/Vis spectrophotometer (OPTIZEN POP, Mecasys, Daejeon, Korea). MPO activity was compared to neutrophil standards, and skin protein levels were determined using the Lowry method [43]. Results were expressed as MPO activity (number of total neutrophils/mg of protein).

#### 4.2.7. Real-Time Polymerase Chain Reaction (RT-PCR)

The mRNA expressions of TGF-β1; p38 MAPK; AKT; COL1A1 and 2; Has 1, 2, and 3; MMP-1, 9, and 13; Nox2, and GSH reductase in the prepared dorsal back skin tissues were evaluated using real-time RT-PCR in accordance with prior studies [13,47]. To elaborate, RNA extraction was carried out using Trizol reagent (Invitrogen, Carlsbad, CA, USA), and RNA concentration and quality were determined using the CFX96^TM^ Real-Time System (Bio-Rad, Hercules, CA, USA). To eliminate any DNA contamination, samples underwent treatment with recombinant DNase I (DNA-free; DNA-free DNA removal kit; Cat No. AM1906, Thermo Fisher Scientific Inc., Rockford, IL, USA). RNA was then reverse transcribed using the High-Capacity cDNA Reverse Transcription Kit (Cat No. 4368813, Thermo Fisher Scientific Inc., Rockford, IL, USA) in accordance with the manufacturer’s instructions. The analysis was performed utilizing the ABI Step One Plus Sequence Detection System (Applied Biosystems, Foster City, CA, USA), and their expression levels were normalized relative to the vehicle control. The following thermal conditions were applied: 10 min at 94 °C, followed by 39 cycles of 15 s at 94 °C, 20 s at 57 °C, and 30 s at 72 °C. The data was normalized using β-actin mRNA expression through the comparative threshold cycle method [49]. The oligonucleotide primer sequences for PCR are provided in Appendix A.

#### 4.2.8. Histopathology

Dorsal back skin samples from the gluteal regions were crossly trimmed and fixed in 10% neutral buffered formalin for 24 h for histopathological observation. Paraffin blocks were created using an automated tissue processor (Shandon Citadel 2000, Thermo Scientific, Waltham, MA, USA) and embedding center (Shandon Histostar, Thermo Scientific, Waltham, MA, USA), and 3~4 μm sections were prepared using automated microtome (RM2255, Leica Biosystems, Nussloch, Germany), followed by preparation of 3 to 4 μm sections through automated microtome. These sections were stained with hematoxylin and eosin (HE) for general histopathology and Masson’s trichrome (MT) for collagen fibers. A histological examination was performed using light microscopy (Model Eclipse 80*i*, Nikon, Tokyo, Japan) with a camera system (ProgRes^TM^ C5, Jenoptik Optical Systems GmbH, Jena, Germany) and image analyzer (*i*Solution FL ver 9.1, IMT *i*-solution Inc., Bernaby, BC, Canada). Various parameters such as epithelial microfolds (folds/mm of epithelium), epithelial thicknesses (μm/epithelium), inflammatory cell count in the dermis (cells/mm^2^ of dermis), and collagen fiber distribution were quantified using computer-assisted image analysis (%/mm^2^ of dermis) according to our previously established methods [13,47]. The histopathologist was unaware of group distribution during analysis. Central regions of samples were selected for observation, resulting in a minimum of one field per dorsal back skin tissue and a total of 10 histological fields per group.

#### 4.2.9. Immunohistochemistry

Immunoreactivities against nitrotyrosine (NT), 4-hydroxynonenal (4-HNE), cleaved caspase-3 and poly (ADP-ribose) polymerase (PARP), and MMP-9 on the dorsal back skin were visualized with specific antibodies (refer to Table 2) using an avidin-biotin-peroxidase complex (ABC) and peroxidase substrate kit (Vector Labs, Burlingame, CA, USA) [38,47]. Endogenous peroxidase activity was blocked by incubating in methanol and 0.3% H_2_O_2_ for 30 min. Non-specific binding of immunoglobulin was prevented by using a normal horse serum blocking solution for 1 h in a humid chamber, following epitope retrieval in 10 mM citrate buffers (pH 6.0) by heating (95–100 °C). Primary antibodies were incubated overnight at 4 °C in a humidity chamber, followed by biotinylated secondary antibodies and ABC reagents. Sections were exposed to the peroxidase substrate kit for 3 min at room temperature. Between each step, all sections were rinsed three times in 0.01 M phosphate buffered saline. Positive immunoreactivity was considered for epithelial cells with over 40% immunoreactivity density for each antiserum in comparison to the background—NT, 4-HNE, cleaved caspase-3, PARP, and MMP-9. The mean numbers of cleaved caspase-3 and PARP, as well as NT and 4-HNE-immunolabeled epithelial cells (% of cells/100 epithelial cells) were quantified using an automated image analysis process and histological camera system. MMP-9 immunoreactive fiber percentages were calculated in the dermis (%/mm^2^). The histopathologist performed the analysis in a blinded manner with respect to group distribution adapted from prior methodologies [38,47].

### 4.3. Statistical Analyses

All collected data were presented as mean ± SD. In vitro and in vivo data were subjected to multiple comparison tests, including the Levene test to assess variance homogeneity. One-way ANOVA followed by Tukey’s HSD test was applied to data with no significant deviations from variance homogeneity, while Dunnett’s T3 test was used for data with significant deviations. Nonparametric comparisons were performed using the Kruskal–Wallis H test and Mann–Whitney u test. Statistical significance was considered for *p*-values < 0.05. Statistical analysis was conducted using SPSS for Windows (Release 27.0).

## 5. Conclusions

In this investigation, we assessed KO’s potential in mitigating UVB-induced skin photoaging. Our experimental outcomes demonstrated that oral administration of KO significantly attenuated UVB-induced wrinkle formation, skin water loss, and collagen degradation. These advantageous effects were attributed to the anti-inflammatory, anti-apoptotic, and antioxidant properties inherent in KO, which exhibited similarity to L-AA (100 mg/kg) at an equivalent oral dose level. With these convincing outcomes, KO stands out as a promising candidate for the development of functional products aimed at preventing skin photoaging when used as a dietary supplement.

## Figures and Tables

**Figure 1 marinedrugs-21-00479-f001:**
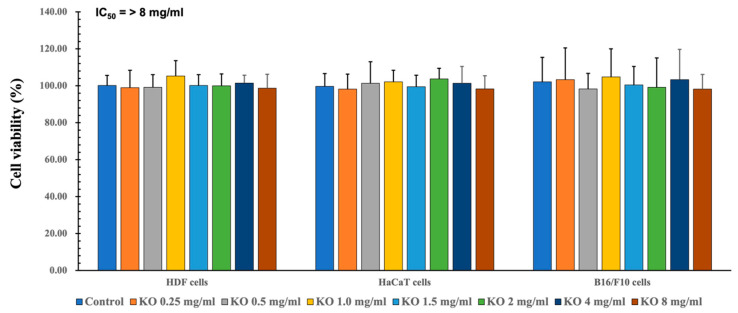
Cytotoxicity of KO on HDF cells, cytotoxicity of KO on HaCaT cells, and cytotoxicity of KO on B16/F10 cells. Data are presented as the mean ± standard deviation (SD). KO, krill oil (Superba^TM^ Boost); HDF, human dermal fibroblasts (neonatal); HaCaT, human keratinocytes; B16/F10, murine melanoma cells.

**Figure 2 marinedrugs-21-00479-f002:**
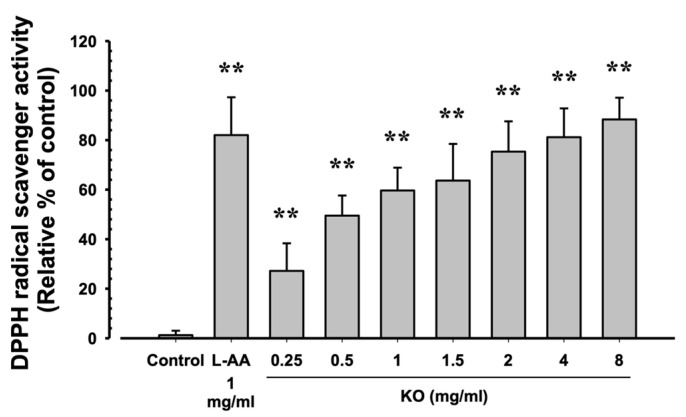
The antioxidant characteristics of KO: Data are presented as the mean ± SD. KO, krill oil (Superba^TM^ Boost); L-AA, L-Ascorbic acid; DPPH, 1-Diphenyl-2-picryhydrazyl radical, 2,2-Diphenyl-1-(2,4,6-trinitrophemyl) hydrazyl. ** *p* < 0.01 as compared with control cells.

**Figure 3 marinedrugs-21-00479-f003:**
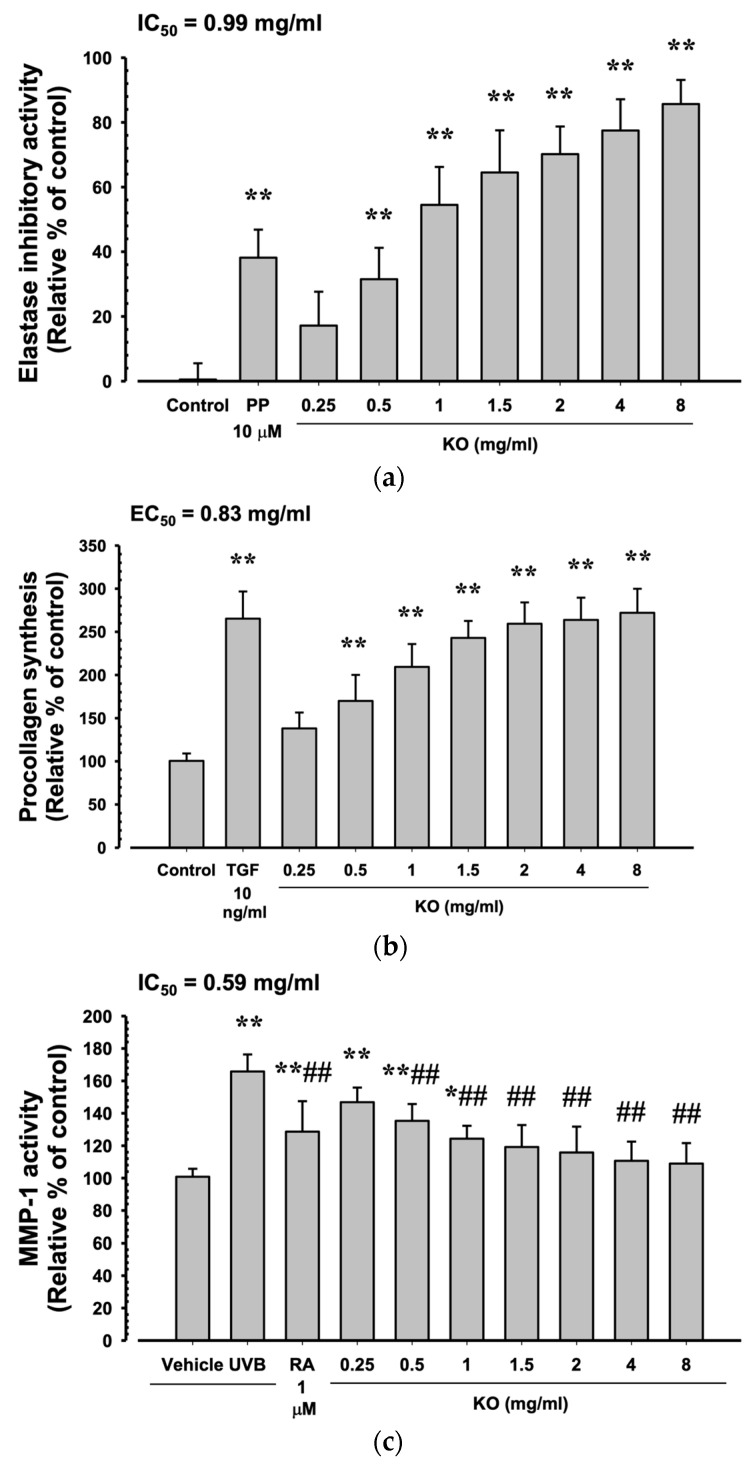
Anti-winkle of KO: (**a**) elastase inhibitory activity; (**b**) collagen synthesis; (**c**) MMP-1 activity. Data are presented as the mean ± SD. KO, Krill oil (Superba^TM^ Boost); PP, phosphoramidon disodium salt; TGF, transforming growth factor; MMP, matrix metalloproteinase; RA, retinoic acid; UVB, ultraviolet B; HDF, human dermal fibroblasts; * *p* < 0.05 and ** *p* < 0.01 as compared with control cells. ^##^
*p* < 0.01 as compared with UVB-irradiated control cells.

**Figure 4 marinedrugs-21-00479-f004:**
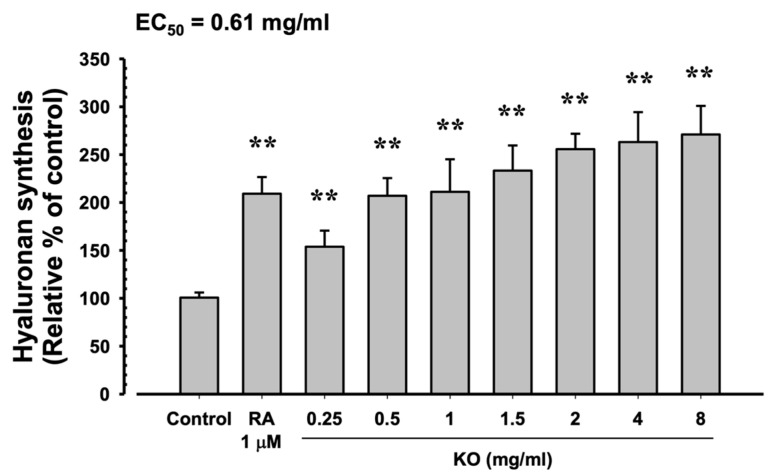
The effects of KO on hyaluronan synthesis: Data are presented as the mean ± SD. KO, krill oil (Superba^TM^ Boost); RA, retinoic acid; HaCaT, human keratinocytes; ** *p* < 0.01 as compared with control cells.

**Figure 5 marinedrugs-21-00479-f005:**
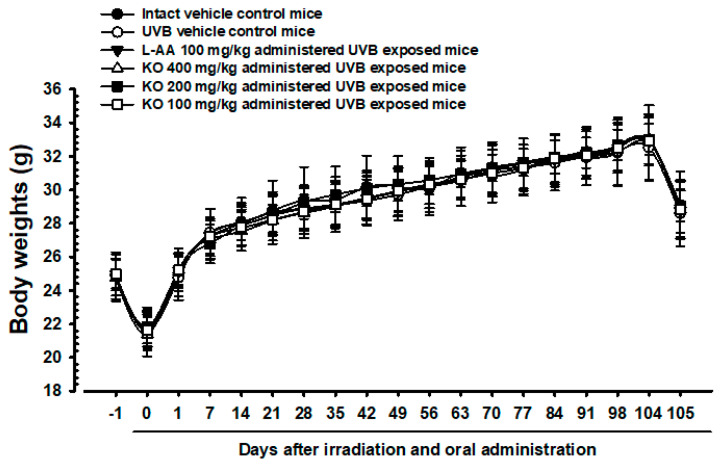
Body weight changes on the days after UVB irradiation and oral administration: KO (100, 200, and 400 mg/kg) or L-AA (100 mg/kg) was orally administrated once a day for 105 days after 1 h of UVB irradiation. The body weights were measured every week. Data are presented as the mean ± SD (*n* = 10, significance compared with intact control mice).

**Figure 6 marinedrugs-21-00479-f006:**
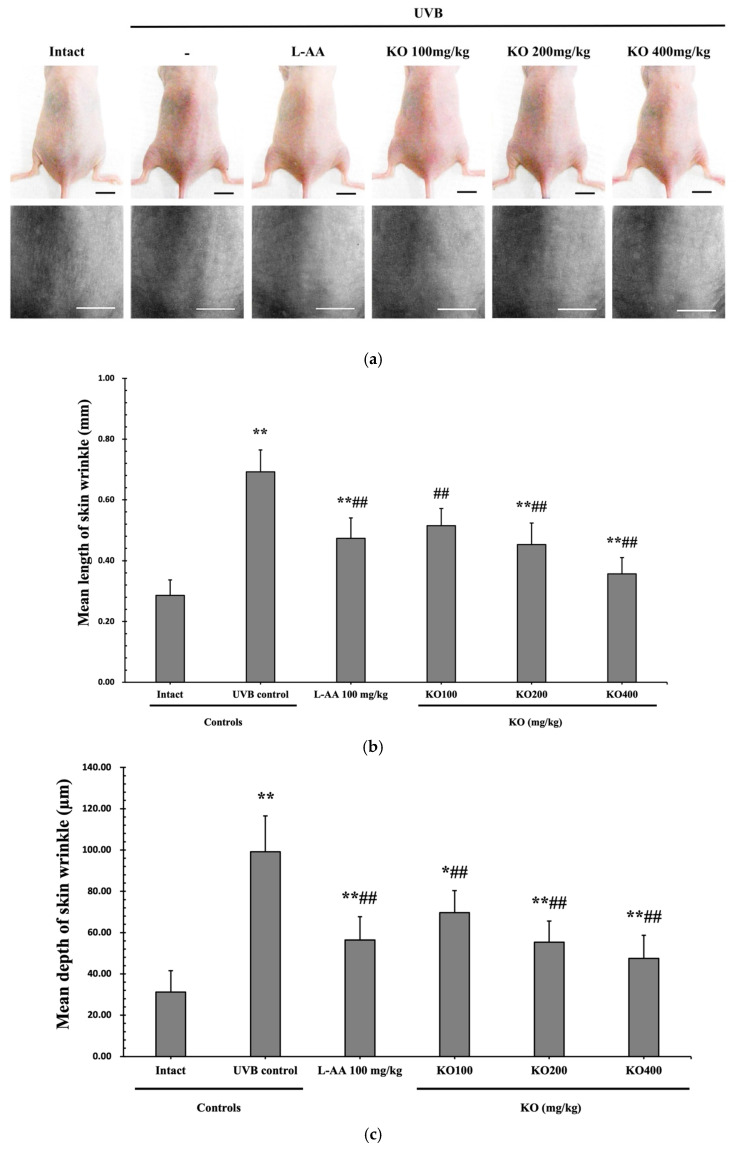
Effects of KO on UVB-induced wrinkle formation in dorsal back skin: (**a**) Photograph of dorsal back skin (**upper**), monochrome image of skin replicas (**lower**). Scale bars indicate 10 mm. Wrinkle shadows were generated using an optic light source by a fixed intensity at a 40° angle.; (**b**,**c**) Wrinkle length and depth; (**d**) Skin water content (6 mm-diameter skin); (**e**) Skin COL1 content (%, relative to intact); (**f**) Skin hyaluronic acid content; (**g**) COL1 synthetic (*COL1A1* and *COL1A2*) in dorsal back skin tissue; (**h**) Hyaluronic acid synthesis (*Has1*, *Has2*, and *Has3*) in dorsal back skin tissue; (**i**) Transforming growth factor *(TGF)-β1 gene* expression in dorsal back skin tissue; (**j**) MMP (*MMP-1*, *MMP-9*, and *MMP13*) gene expression in dorsal back skin tissue. Data are presented as the mean ± SD (*n* = 10, significance difference vs. intact control; * *p* < 0.05, ** *p* < 0.01, vs. UVB-irradiated control mice; ^#^
*p* < 0.05, ^##^
*p* < 0.01).

**Figure 7 marinedrugs-21-00479-f007:**
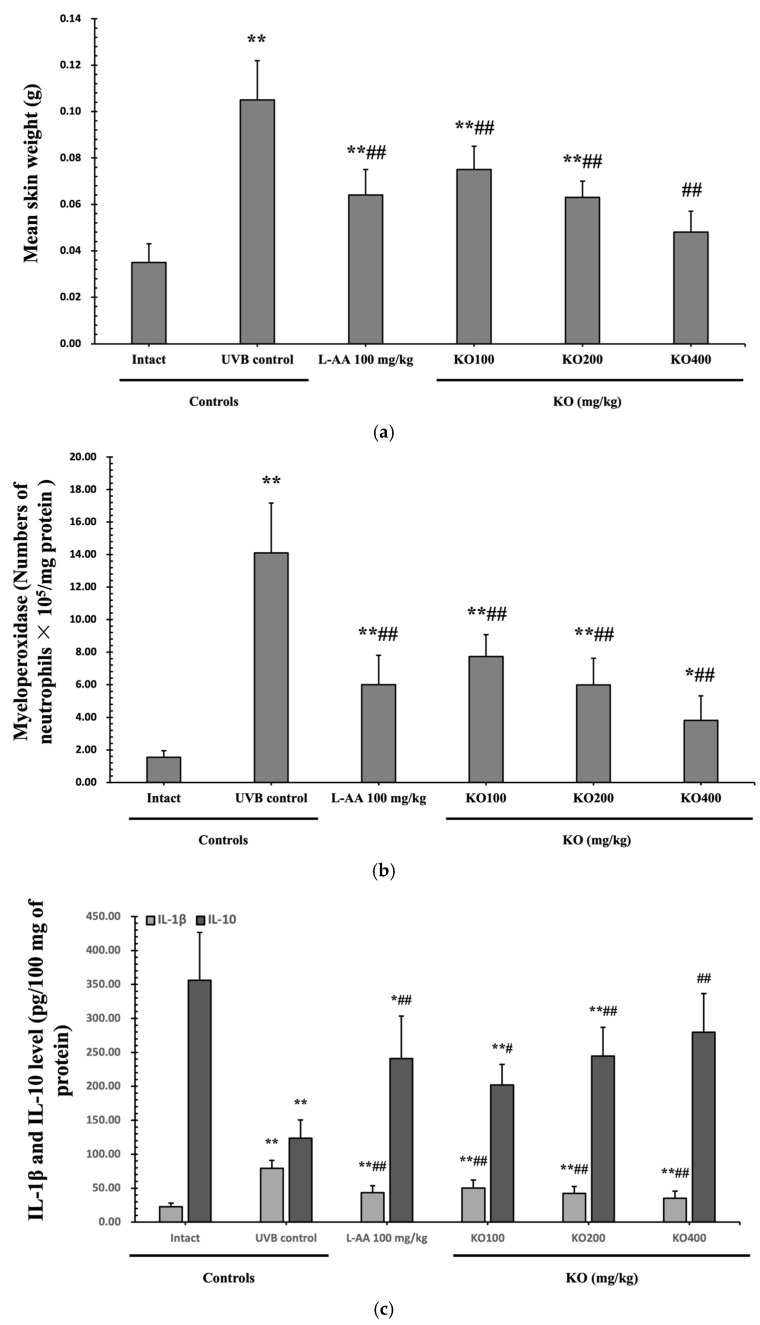
Effects of KO on UVB-induced skin inflammation: (**a**) Skin edema (weight of 6mm diameter skin sample); (**b**) Myeloperoxidase (MPO) activities for skin neutrophil content; (**c**) IL-1β and IL-10 levels in dorsal back skin tissue.; (**d**) p38 mitogen-activated protein kinase (p38 MAPK) gene expression in dorsal back skin tissue; (**e**) protein kinase B (AKT) gene expression. Data are presented as the mean ± SD (*n* = 10, significance difference vs. intact control; * *p* < 0.05, ** *p* < 0.01, vs. UVB-irradiated control mice; ^#^
*p* < 0.05, ^##^
*p* < 0.01).

**Figure 8 marinedrugs-21-00479-f008:**
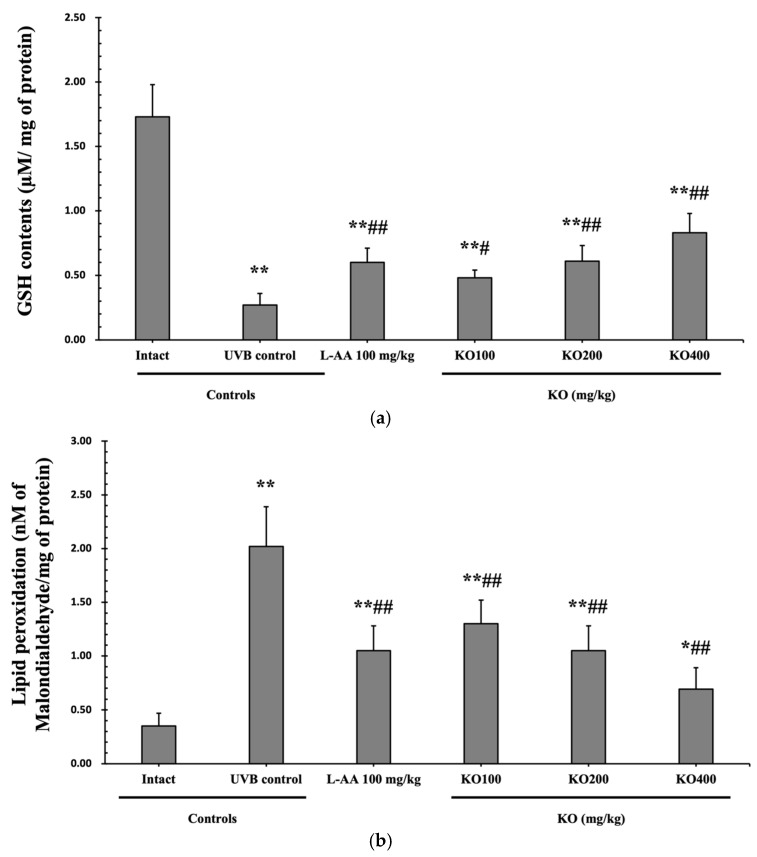
Effects of KO on UVB-induced oxidative stress: (**a**) GSH contents in the skin tissue; (**b**) MDA level in the skin tissue; (**c**) Superoxide anion production in the skin tissue; (**d**) *GSH reductase mRNA* expression level in the dorsal back skin tissue; (**e**) *NOX2 mRNA* expression level in the dorsal back skin tissue. Data are presented as the mean ± SD (*n* = 10, significance difference vs. intact control; * *p* < 0.05, ** *p* < 0.01, vs. UVB-irradiated control mice; ^#^
*p* < 0.05, ^##^
*p* < 0.01).

**Figure 9 marinedrugs-21-00479-f009:**
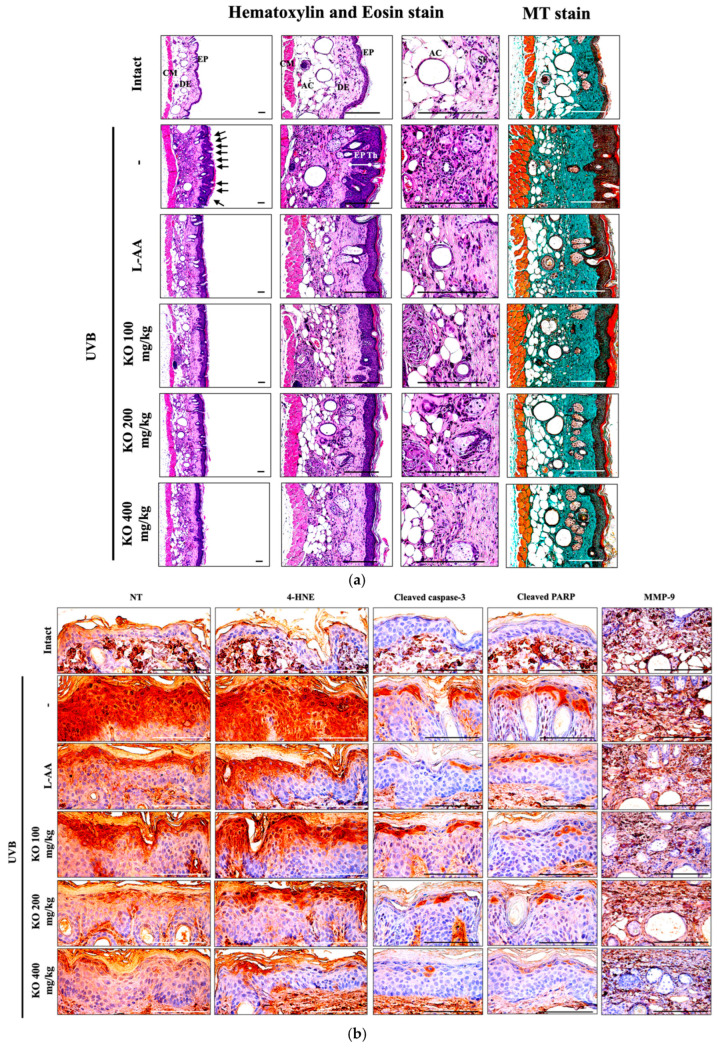
(**a**) Representative images of stained skin with tissue with hematoxylin and eosin or Masson’s trichrome (MT). Arrows indicate microfolds in skin epithelial surface. Scale bars indicate 200 µm. (**b**) Immuno-stained skin tissue using nitrotyrosine (NT), 4-hydroxynonenal (4-HNE), cleaved caspase-3, cleaved PARP, and MMP9 antibodies. Scale bars indicate 100 µm. EP, epithelium; DE, dermis; CM, cutaneous muscle; SE, sebaceous gland; AC, adipocyte; Th, thickness.

**Table 1 marinedrugs-21-00479-t001:** General histomorphometrical analysis of dorsal back skin, taken from unexposed intact or UVB-exposed hairless mice.

Items (Unit) Groups	Number of Microfolds (Folds/mm of Epidermis)	Mean Epithelial Thickness (μm/Epidermis)	Mean Inflammatory Cells (cells/mm^2^ of Dermis)	Collagen Fiber Occupied Regions (%/mm^2^ of Dermis)
Controls				
Intact	3.30 ± 1.06	24.45 ± 4.21	43.40 ± 11.12	31.76 ± 6.58
UVB	17.60 ± 2.37 **	120.25 ± 13.37 **	525.20 ± 118.05 **	75.73 ± 7.50 **
Reference				
L-AA 100 mg/kg	8.90 ± 1.79 **^##^	65.94 ± 10.15 **^##^	188.60 ± 74.90 **^##^	51.47 ± 8.96 **^##^
Test materials				
KO 100 mg/kg	12.10 ± 1.60 **^##^	77.69 ± 10.98 **^##^	242.80 ± 67.05 **^##^	56.78 ± 5.30 **^##^
KO 200 mg/kg	8.10 ± 1.45 **^##^	65.03 ± 11.28 **^##^	186.40 ± 82.58 **^##^	51.43 ± 7.23 **^##^
KO 400 mg/kg	6.10 ± 1.29 **^##^	55.00 ± 10.87 **^##^	134.20 ± 38.05 **^##^	42.95 ± 8.00 *^##^

Significance difference vs. intact control; * *p* < 0.05, ** *p* < 0.01, vs. UVB-irradiated control mice; ^##^
*p* < 0.01).

**Table 2 marinedrugs-21-00479-t002:** Immunohistomorphometrical analysis of dorsal back skin, taken from unexposed intact or UVB-exposed hairless mice.

Groups Items	Controls		Reference	Test Materials		
	Intact	UVB	L-AA 100 mg/kg	KO 100 mg/kg	KO 200 mg/kg	KO 400 mg/kg
Epidermis (cells/100 epithelial cells)						
Nitrotyrosine	18.20 ± 4.47	80.20 ± 7.80 **	41.40 ± 10.98 **^##^	53.60 ± 10.45 **^##^	40.80 ± 11.52 **^##^	30.20 ± 10.69 ^##^
4-HNE	14.00 ± 2.49	86.20 ± 4.94 **	52.00 ± 6.86 **^##^	63.00 ± 10.17 *^##^	51.20 ± 10.92 **^##^	28.00 ± 7.60 **^##^
Cleaved caspase-3	4.60 ± 1.90	36.60 ± 4.22 **	18.40 ± 5.40 **^##^	24.80 ± 4.12 **^##^	18.00 ± 5.25 **^##^	10.00 ± 4.32 ^##^
Cleaved PARP	4.60 ± 1.90	41.00 ± 5.68 **	22.00 ± 3.77 **^##^	27.00 ± 3.68 **^##^	21.00 ± 3.92 **^##^	12.20 ± 2.20 **^##^
Dermis (%/mm^2^)						
MMP-9	20.60 ± 5.17	70.14 ± 7.05 **	43.50 ± 9.50 **^##^	53.95 ± 8.33 **^##^	41.98 ± 10.77 **^##^	32.75 ± 9.27 *^##^

Data are presented as the mean ± SD (*n* = 10, significance difference vs. intact control; *****
*p* < 0.05, ******
*p* < 0.01, vs. UVB-irradiated control mice; ^##^
*p* < 0.01). 4-HNE, 4-Hydroxynonenal; PARP, Poly(ADP-ribose) polymerase; MMP, matrix metalloprotease.

## Data Availability

Data are contained within the article.

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
