# Peer review of "Krill Oil’s Protective Benefits against Ultraviolet B-Induced Skin Photoaging in Hairless Mice and In Vitro Experiments"

_marinedrugs, 2023, doi:10.3390/md21090479_

Round 1
Reviewer 1 Report
This research article presents the Krill Oil's Protective Benefits Against Ultraviolet B-Induced 2 Skin Photoaging in Hairless Mice and In Vitro Experiments. It appears to be an interesting topic. Overall, the manuscript is well-conceived, undertaken, written and presented. Minor revisions of content are required prior to acceptance for publication.
Abstract:- Line 14: Provide the assays/experiments that have been conducted for in vitro cell models as well.
- Line 16: Why 66 hairless mice? Should not a total of 6 groups, with 10 mice per group?
- Emphasize the results of cytotoxicity effect and no body weight changes by KO
Introduction: introduce/ justify the 3 cell lines (HaCaT, HDF, and B16/F10 cells) selected for this study
- Line 77: Suggest combine the results of cytotoxicity of KO in HaCaT, HDF, and B16/F10 cells in a graph.
- Line 89: Adjust the font size
- Line 102 & 106: Provide full term for PP and TGF-beta
- Line 298: for Fig 9(a), it should be MT stain but not MTT stain.
- The methodology to study the effect of KO on body weight changes is not provided in Section of Materials and Methods
No major issues.
Author Response
We have provided responses and explanations in the file we uploaded.

Reviewer 2 Report
Comments
1. In line 36 the author mentions “Unlike the mechanism of UVA-induced pigmentation”, what does this serve to highlight?
2. The Materials and Methods section of the article is not described clearly enough, e.g. the description of 4.1.4.
3. In lines 566-568, it is mentioned that “RT-PCR was employed to detect variations in the mRNA expression of TGF-β1, p38 MAPK, AKT, MMP-1, MMP-9, and MMP-13.” but there is nothing in the article about TGF-β1, p38 MAPK, AKT assay results.
4. In Figure 5, what is the reason for the decrease in body weight of the mice on the first day of administration?
5. What was the purpose of the article to determine the cytotoxicity of KO on B16/F10 cells? The connection of this to other experiments is not well represented in the article.
6. In lines 413-415, “ Our study found that UVB exposure and aging downregulated the genes responsible for hyaluronic acid synthesis (HAS1, HAS2, and HAS3) in the dermis.” why are the values for the intact and UVB control groups so close in Figure 6h? Same question for Figure 6g.
7. The picture in Figure 6a is not clear enough.
8. Lines 88-90, “Cytotoxicity of KO on HDF (a), HaCaT cell (b), and B16F10 (c) B16/F10 Cells:” please examine this sentence carefully.
9. In line 106, “(Figure 3A)” and in line 110 “(Figure 3B)” should read “(Figure 3a)” and “(Figure 3b)”.
Author Response

(The authors gave the same response as above.)

Reviewer 3 Report
Dear Authors,
I write you in regard to your manuscript entitled Krill Oil's Protective Benefits Against Ultraviolet B-Induced Skin Photoaging in Hairless Mice and In Vitro Experiments. The investigation was well justified and text was adequately written, in terms of structure.
- it was not completely clear the objectives when the ascorbic acid was involved. Abstract, for example, did not mention this sample.
- did the KO develop better antioxidant activity in comparison with the ascorbic acid? This response was not described in the Results.
- the use of animals of experimentation in the cosmetic is highly criticized. why did authors use such model? If the study motivation was the use of KO as a food supplement, it wild partially justify the research, but if the topical use was the future motivation, this model must be revised.
- overall, the title and text must explicit that the KO use was as a food or oral to avoid any misinterpretation with topical application.
Author Response

(The authors gave the same response as above.)

Round 2
Reviewer 3 Report
Dear Authors,
Thank you for addressing all questions from the peer-review process. On last comment, perhaps, could be considered. In conclusion, the term therapeutic could not be appropriate for the KO.